# Human Skin Aging and the Anti-Aging Properties of Retinol

**DOI:** 10.3390/biom13111614

**Published:** 2023-11-04

**Authors:** Taihao Quan

**Affiliations:** Department of Dermatology, University of Michigan Medical School, Ann Arbor, MI 48109, USA; thquan@umich.edu

**Keywords:** skin aging, retinol, collagen, MMPs, TGF-β

## Abstract

The skin is the most-extensive and -abundant tissue in the human body. Like many organs, as we age, human skin experiences gradual atrophy in both the epidermis and dermis. This can be primarily attributed to the diminishing population of epidermal stem cells and the reduction in collagen, which is the primary structural protein in the human body. The alterations occurring in the epidermis and dermis due to the aging process result in disruptions to the structure and functionality of the skin. This creates a microenvironment conducive to age-related skin conditions such as a compromised skin barrier, slowed wound healing, and the onset of skin cancer. This review emphasizes the recent molecular discoveries related to skin aging and evaluates preventive approaches, such as the use of topical retinoids. Topical retinoids have demonstrated promise in enhancing skin texture, diminishing fine lines, and augmenting the thickness of both the epidermal and dermal layers.

## 1. Introduction

The passage of time is an inevitable aspect of life, and with it comes the process of aging. In humans, aging is a natural progression that involves both physical and psychological changes. Physically, the human body undergoes numerous transformations over time. Cells and tissues gradually deteriorate, leading to a decline in the body’s ability to function optimally. Organs may become less efficient, and the body may experience a decrease in strength, flexibility, and overall vitality. The physical signs of aging primarily appear on the skin, such as wrinkles, gray hair, and changes in skin texture. These indications tend to become increasingly apparent with advancing age. The visible and conspicuous signs of aging that manifest on our skin hold considerable importance in our social interactions and visual appearance. Consequently, the condition and appearance of our skin can profoundly influence our emotional and psychological well-being, consequently affecting our overall quality of life.

The skin serves a critical role as the primary defense barrier against various environmental threats, including exposure to solar ultraviolet (UV) radiation, physical and chemical injuries, infections from pathogens, and the prevention of water loss. Human skin comprises two primary layers: the outer layer, referred to as the epidermis, and the underlying layer, known as the dermis. The epidermis is primarily composed of keratinocytes, which produce keratins and form the stratum corneum, the outer protective layer of the skin. In contrast, the dermis contains fewer cells and consists mainly of proteins such as collagen, elastin, fibronectin, and proteoglycans, which constitute the extracellular matrix (ECM). Collagen, making up approximately 90% of the skin’s dry weight, stands as the predominant protein. Dermal fibroblasts play a crucial role in synthesizing, organizing, and maintaining the collagen-rich ECM within human skin.

Like all other organs, human skin undergoes a natural aging process as time passes. However, unlike many other organs, human skin is continually exposed to environmental stresses and damage, particularly from sources such as solar UV radiation [1]. The aging of the skin can be categorized into two distinct types based on their underlying causes: intrinsic or chronological aging, which occurs as a natural consequence of the passage of time, and extrinsic aging, often referred to as photoaging, which results from external factors, most notably UV radiation exposure [1]. Chronological or intrinsic aging encompasses changes that occur in everyone due to the natural passage of time, while photoaging or extrinsic aging is linked to repeated exposure to UV radiation from the Sun. Both intrinsic and extrinsic aging are linked to a reduction in collagen production and an elevation in collagen breakdown, albeit with differing underlying mechanisms. Consequently, both intrinsic and extrinsic aging share common molecular characteristics.

Over time, both forms of aging accumulate gradually, with photoaging exerting its influence on the skin alongside the changes brought about by chronological aging. The regions of the face, neck, forearms, and lower legs display the most-noticeable signs of aging, as they undergo a combination of both chronological aging and photoaging [1]. These areas are also where skin diseases related to aging are most commonly observed. Histological and ultrastructural studies provide compelling evidence that the morphology of human skin undergoes significant alterations during the aging process. A prominent feature of aging in older individuals is the thinning of both the epidermal and dermal layers [2]. Multiple lines of evidence strongly support the notion that the age-related thinning of the epidermis has a substantial impact on skin function. This thinning notably compromises the skin’s ability to function as an effective protective barrier against environmental aggressors and to retain moisture, resulting in increased moisture loss. In addition to the epidermis, the dermal layer of aged skin also experiences thinning, primarily attributed to the loss of collagen, which serves as the skin’s primary structural protein. These age-related changes in the epidermis and dermis directly contribute to various skin conditions, including heightened fragility [3], compromised vascular support [4], delayed wound healing [5,6], and an increased susceptibility to cancer development [7,8,9]. Consequently, skin aging extends beyond mere cosmetic concerns, as it significantly contributes to age-related skin issues.

## 2. Molecular Mechanisms of Human Skin Aging

Intrinsic skin aging is triggered by a variety of factors, which include genomic instability, cellular senescence, and the shortening of telomeres. On the other hand, extrinsic aging is accelerated by external factors such as UV radiation, pollution, tobacco smoke, and alcohol consumption. These external elements expedite the aging process, leading to premature skin aging. As the aging process unfolds, the skin undergoes a series of changes, often characterized by the thinning and flattening of the epidermis, along with damage to the dermis due to the breakdown of collagen fibrils. Additionally, an inflammatory environment referred to as “inflammaging” becomes apparent [2].

### 2.1. Epidermal Aging

The epidermis, primarily composed of keratinocytes, functions as a protective skin layer that serves as a barrier between the body and the surrounding environment. With the passage of time, a series of alterations transpire within the epidermis, collectively denoted as epidermal aging. These modifications are characterized by the thinning of the epidermal layer and the flattening of rete ridges (as depicted in Figure 1, on the right). The principal cause of epidermal aging can be traced to a reduction in the proliferation and turnover of keratinocytes, primarily linked to the depletion of interfollicular epidermal (IFE) stem cells.

Despite the broad recognition of epidermal aging, the mechanisms responsible for this phenomenon remain elusive. COL17A1 has been of particular interest due to its role in maintaining the homeostasis of the skin stem cells [2,10,11]. COL17A1 serves as a structural element within the dermal–epidermal basement membrane [12], and it is synthesized by epidermal keratinocytes, not fibroblasts [13]. COL17A1 exhibits its primary expression in the uppermost extensions of the rete ridges area, where the niches for IFE stem cells are located (Figure 1). Research results have suggested a reduction in the expression of COL17A1 in human skin affected by both intrinsic and extrinsic photoaging, as well as in human skin exposed to acute UV radiation [13,14]. The decrease in COL17A1 levels within the area specific to the rete ridges can potentially diminish the adherence of IFE stem cells to their designated locations, leading to their removal from the skin [13,15]. Consequently, the deficiency of COL17A1 results in decreased rates of keratinocyte renewal and the development of thinner epidermal layers, which constitute the primary morphological characteristic of aging skin. Decreased COL17A1 expression also contributes significantly to the aging process of hair [10,16], which is a natural and unavoidable process as individuals grow older. The most-prominent signs of hair aging are the graying of hair and a decrease in hair density. Gray hair primarily occurs because of the depletion of melanocyte stem cells (McSCs), whereas hair loss is a consequence of diminished hair follicle stem cells (HFSCs). COL17A1 is located in the basal membrane of hair follicles and can impact the maintenance of homeostasis in hair follicle stem cells. The age-related decrease in COL17A1 expression directly affects the integrity of the basement membrane, potentially disrupting the homeostasis of hair follicle stem cells and contributing to hair aging. The absence of COL17A1 can also weaken the junction between the epidermis and dermis, leading to various age-related skin changes, such as heightened skin fragility, blistering, and delayed wound healing. Moreover, the weakened junction between the epidermis and dermis can result in the formation of wrinkles and sagging skin. Additionally, COL17A1 has been linked to age-related skin conditions like bullous pemphigoid, an autoimmune blistering disease [12]. Ongoing research is examining the role of COL17A1 in epidermal aging, and further investigations are necessary to fully understand the underlying mechanisms and to explore potential interventions or treatments.

### 2.2. Dermal Aging

The largest portion of the skin’s volume is taken up by the dermis, and it has a crucial function in supplying mechanical strength and physical durability. The dermis mainly consists of tightly woven collections of collagen fibrils, creating an intricate three-dimensional ECM. Dermal fibroblasts, residing within and around the collagen bundles, are responsible for producing this collagenous ECM. As we grow older, there are adverse alterations that occur in both collagen bundles and fibroblasts. These alterations encompass the fragmentation and disarray of collagen fibrils, decreased collagen synthesis, and the establishment of an inflammatory microenvironment within the dermis, commonly referred to as “inflammaging” [2,17]. These age-related alterations in the dermal microenvironment directly contribute to various skin conditions, including increased fragility [3], impaired support for blood vessels [4], compromised wound healing [5,6], and the promotion of cancer development [2,7,8]. As outlined below, the aging process in the dermis can mainly be attributed to three specific molecular events: (1) the breakdown of collagen fibers due to the actions of matrix metalloproteinases (MMPs), (2) reduced collagen production caused by compromised TGF-β signaling, and (3) the presence of an inflammatory microenvironment known as “inflammaging” (Figure 1, right).

#### 2.2.1. Dermis Damage Caused by the Age-Related Increase in MMPs Leading to Collagen Fragmentation

The fragmentation, lack of organization, and reduction in the collagen-rich ECM are distinctive features of aged human skin [1]. MMPs constitute a class of proteases that play a specific role in breaking down ECM proteins. Presently, the human MMP gene family comprises more than 20 members, each displaying unique structures and preferences for substrates [18]. These MMPs are implicated in a variety of natural and pathological processes linked to ECM degradation. In human skin, MMP1, also known as collagenase 1, is predominantly synthesized by dermal fibroblasts [19]. MMP1 serves as a crucial enzyme responsible for initiating the breakdown of the original collagen fibrils. In young skin, the levels of MMP1 are minimal; however, a significant increase in aged skin dermis is experienced [19,20]. Alongside MMP1, several other MMPs also show heightened levels in the aged dermis [20]. The MMP-mediated fragmentation of the collagen-rich ECM leads to irreversible disruption of the dermis’s structural and mechanical integrity. This disruption has detrimental effects on the skin’s ability to resist bruising and impacts the functional interactions between dermal cells and the ECM. These detrimental changes are believed to significantly contribute to several noticeable clinical characteristics of aging skin, such as thinning, increased fragility, compromised wound healing, and an elevated risk of developing carcinoma.

#### 2.2.2. Dermis Thinning Due to Age-Related Impairment of TGF-β Signaling, Leading to Decreased Collagen Production

Collagen plays a crucial role in maintaining the skin’s structural integrity and strength. The diminished synthesis of collagen stands as a key factor contributing to the thinning of the dermis in aging skin [2]. Transforming growth factor-beta (TGF-β), a versatile cytokine, controls various cellular activities such as cell growth and differentiation, and serves as a primary regulator of ECM components like collagen and elastin [21]. In aged human skin, there is a decrease in the expression of the TGF-β Type II receptor (TβRII) within dermal fibroblasts, leading to impaired TGF-β signaling [22]. This impairment in TGF-β signaling can significantly impact the aging process of the skin by inhibiting collagen production. Diminished TGF-β signaling can result in the increased activity of MMPs, which subsequently break down collagen and other ECM proteins. Aberrant TGF-β signaling can also play a part in inflammaging, a phenomenon recognized for hastening the aging process. Overall, impaired TGF-β signaling can result in decreased skin elasticity, firmness, and resilience. This can manifest as sagging, wrinkles, fine lines, and an overall deterioration in skin quality, all of which are characteristic traits of skin dermal aging. Strategies aimed at addressing impaired TGF-β signaling and its consequences on skin aging may involve targeted therapies and interventions designed to boost collagen production, regulate the ECM, and mitigate skin inflammation.

#### 2.2.3. The Impact of CCN1 Protein on the Aging Process of Human Skin

The role of the CCN1 protein (also known as CYR61) in human skin aging has gained significant attention in recent years [22]. CCN1 is the first member of the CCN protein family, which has six members designated as CCN1 to CCN6 [23,24,25]. CCN1 has been reported to regulate cell adhesion and migration, chemotaxis, the production of inflammatory mediators, cell–matrix interactions, the synthesis of ECM proteins, and wound responses, in a variety of cells in culture and animal models [23,26,27]. Elevated CCN1 expression can lead to a cellular phenotype characterized by enhanced secretion of cytokines, chemokines, growth factors, and proteases [28,29,30,31].

In human skin, CCN1 is predominantly expressed in dermal fibroblasts and significantly elevated in aged human skin [32]. Remarkably, increased CCN1 expression by dermal fibroblasts has a detrimental impact on the dermal microenvironment by promoting the senescence-associated secretory phenotype (SASP) [33,34]. The CCN1 protein is emerging as a pivotal player in human skin aging, influencing collagen production, oxidative stress, and inflammation [22]. Elevated CCN1 contributes to skin dermal aging through several distinct mechanisms [22,32]: it reduces the expression of dermal collagen by inhibiting the TGF-β signaling pathway, resulting in a thinner dermis; it increases the expression of multiple transcription factors via Activator Protein 1 (AP-1), leading to a damaged dermis; it enhances proinflammatory cytokine expression by inducing the generation of reactive oxygen species (ROSs), leading to inflammaged skin.

As such, CCN1-induced alterations of the dermal microenvironment account for many of the characteristic features of tagged human skin. A comprehensive understanding of CCN1’s functions in the skin offers valuable insights into the mechanisms of skin aging. Further research is needed to explore the therapeutic potential of targeting CCN1, which may provide innovative strategies for addressing age-related skin concerns and maintaining healthier, more-youthful skin.

#### 2.2.4. Inflammaging Due to Increased Expression of Multiple Cytokines in Aged Human Skin

Inflammaging is a term coined by combining “inflammation” and “aging”, describing a persistent, low-grade inflammatory state, which tends to intensify as people grow older [35]. This concept is rooted in the fields of gerontology and immunology, highlighting the phenomenon where the immune system becomes increasingly active, resulting in more-frequent and sustained inflammatory responses as individuals age. This inflammatory state can affect various tissues and organs throughout the body, often manifesting as a systemic condition with widespread impacts. Elevated levels of pro-inflammatory markers like interleukin-6 (IL-6), tumor necrosis factor-alpha (TNF-α), and C-reactive protein (CRP) are frequently observed in older individuals experiencing inflammaging. IL-6 has become a widely adopted indicator of inflammaging and is considered a defining characteristic of chronic illness [36].

Notably, studies have indicated an increase in these inflammaging-related cytokines in the skin of aged individuals in vivo [33], which can contribute to skin aging. Inflammaging accelerates skin aging by promoting inflammation, breaking down collagen, and inhibiting collagen production in the skin. While the exact sources of these inflammaging-related cytokines are not fully understood, they can originate from various cell types within the body, including immune cells and various tissues. Considering that the skin is the body’s largest organ and is frequently subjected to stresses from both internal factors (oxidative stress) and external factors (UV irradiation), it is conceivable that the skin plays a significant role in producing inflammaging-related cytokines [37,38]. The activator protein-1 (AP-1) transcription factor plays a pivotal role in the expression of numerous cytokines. The activation and induction of the AP-1 transcription factor primarily occur through the mitogen-activated protein kinase (MAPK) pathways. The activation of MAP kinases triggers the expression of both c-Jun and c-Fos, which are the major components of the AP-1 transcription factor. Interestingly, research has shown that c-Jun is upregulated in aged human skin, while c-Fos is consistently expressed [39], suggesting that increased c-Jun activity may be a key driver behind the heightened production of inflammaging-related cytokines in the skin of elderly individuals.

Inflammaging negatively affects the age-related decline in skin barrier function [37,40]. Inflammaging disrupts the epidermal lipid structure, rendering it more vulnerable to dehydration and external stressors. Interestingly, a study by Hu et al. demonstrated that disrupting the epidermal barrier function in young mice led to elevated levels of proinflammatory cytokines in both the skin and serum [38,41]. Conversely, restoring the epidermal barrier function in aged mouse skin resulted in reduced circulating cytokine levels. These findings imply that the epidermis could have a significant role in the age-related systemic inflammation observed in inflammaging. As the source of systemic inflammatory cytokines in inflammaging remains elusive, it is intriguing to explore whether enhancing epidermal barrier function could similarly alleviate inflammaging in humans. The results of such research could provide a potential pathway to mitigate inflammaging and, subsequently, reduce the risk of age-related diseases. In this context, the study conducted by Hu et al. serves as a proof of concept, indicating that preserving epidermal barrier function is an effective approach to reducing systemic inflammation as individuals age. Understanding the mechanisms underlying this phenomenon is pivotal for developing strategies to counteract its effects. Ongoing research in this field continually furnishes valuable insights into the connections between inflammation and skin aging, offering novel possibilities for potential interventions and treatments.

#### 2.2.5. Autophagy and Skin Aging

Autophagy is a crucial cellular process that maintains homeostasis by eliminating damaged organelles and protein aggregates or recycles cytoplasmic components through lysosomes [42]. Three distinct types of autophagy exist: macroautophagy, microautophagy, and chaperone-mediated autophagy. In the process of organismal aging, autophagy and various other lysosomal transport pathways demonstrate diminished activity [43,44,45]. The skin, being an organ with limited access to nutrients, relies on autophagy to preserve its scarce resources and maintain homeostasis. Recent research has demonstrated the critical role of autophagy in maintaining skin homeostasis, and the dysregulation of autophagic processes have been associated with age-related skin conditions [46,47]. Skin stem cells, melanocytes, Merkel cells, and sweat gland secretory cells all rely on autophagy to maintain their homeostasis [48].

In the dermis, autophagy has been linked to age-related cellular senescence and alterations in the collagenous ECM [48,49,50]. The role of autophagy in the process of dermal aging is a subject of significant interest in the fields of skin biology and skin aging [51,52]. As skin undergoes the aging process, the effectiveness of autophagy tends to diminish. Consequently, cells become less capable of efficiently removing damaged or malfunctioning cellular components. This decline in autophagic activity results in the buildup of cellular waste, including misfolded proteins and impaired mitochondria. This accumulation, in turn, can trigger oxidative stress and inflammation, which are characteristic features of aging skin. Autophagy also plays a pivotal role in the breakdown of collagen and elastic fibers within the dermal layer of the skin. A reduction in autophagy can lead to the accumulation of damaged collagen and elastin, contributing to the development of sagging skin and wrinkles.

The decline in proteostasis, which refers to the maintenance of protein quality control and integrity, as a consequence of aging, has been reported [53]. When the ability to clear oxidized and misfolded proteins diminishes, it can trigger the activation of DNA damage repair (DDR) pathways, subsequently leading to the initiation of cellular senescence [54] and the secretion of senescence-associated secretory phenotype (SASP) factors [55].

Autophagy in fibroblasts is essential for eliminating lipofuscin, which is an accumulation of proteins and lipids that have become misfolded and modified [56]. The compromised autophagic activity observed in senescent human fibroblasts [57] could be linked to the abnormal buildup of lipofuscin, which is known to be the root cause of age-related pigmentation irregularities.

Autophagy also plays a role in maintaining the integrity of the epidermal protective barrier by regulating the regeneration of skin cells, especially those located in the outermost layer of the epidermis [46,58]. The reduced autophagic activity associated with aging may lead to a weakened skin barrier, rendering the skin more susceptible to environmental stressors and greater moisture loss.

One of the most well-understood aspects of autophagy in skin aging is mitophagy [59], a form of selective autophagy that targets damaged mitochondria. As we age, the efficiency of mitophagy may decline, leading to the accumulation of dysfunctional mitochondria, which further exacerbates skin aging.

Several key regulators of autophagy in skin cells have been identified, including the mammalian target of rapamycin (mTOR), AMP-activated protein kinase (AMPK), and autophagy-related (ATG) genes [43,60,61]. These molecules control the initiation and progression of autophagy, making them attractive targets for potential interventions to combat aging. Pharmaceutical agents targeting autophagy regulators like mTOR and AMPK are under investigation for their potential to promote skin health and slow the aging process [62].

Autophagy in the skin is a vital process that maintains skin homeostasis. Its decline with age contributes to various aspects of skin aging, including the breakdown of structural components, diminished barrier function, and increased susceptibility to environmental damage. Exploring strategies to boost autophagy in the skin presents exciting possibilities for mitigating the signs of aging and promoting overall skin health.

#### 2.2.6. A Mouse Model of Dermal Aging Achieved through the Fibroblast-Specific Expression of Human MMP1

Two mouse models of dermal aging were recently reported by the selective expression of human MMP1 [9] or human CCN1 [63] in dermal fibroblasts driven by the fibroblast-specific collagen1A2 (*Col1a2*) promoter and upstream enhancer [64,65]. These mouse models of dermal aging are based on the human skin in vivo observations that both MMP1 and CCN1 are significantly elevated in dermal fibroblasts of aged human skin and drive dermal aging by establishing an aberrant dermal ECM microenvironment [22]. At six months of age, both *Col1a2;hMMP1* [9] and *Col1a2;hCCN1* [63] mice displayed many of the hallmarks of aged human skin dermis. Importantly, *Col1a2;hMMP1* and *Col1a2-hCCN1* mice exhibit an enhanced preponderance of keratinocyte cancer in mouse models of skin carcinogenesis: cutaneous two-stage chemical carcinogenesis and inducible HRas-oncogene-driven tumors (Figure 2).

A noteworthy discovery in this dermal aging mouse model was the impaired spreading and morphology of fibroblasts in an environment with fragmented collagen fibrils. Dermal fibroblasts are naturally situated within the collagen-rich ECM microenvironment, and their direct interactions with collagen fibrils allow them to generate mechanical forces that govern their shape and function. In aging skin, the breakdown of collagen fibrils does not offer sufficient support for fibroblast attachment, leading to decreased spreading of fibroblasts and diminished mechanical strength. In this state, fibroblasts take on an aged phenotype, perpetuating collagen degradation, reduced collagen production, and the emergence of an inflammaging milieu [66]. These findings were substantiated in hMMP1 transgenic mice, where the loss of fibroblast spreading and an altered morphology led to decreased collagen production, downregulated TβRII expression, upregulated expression of several MMPs, and an increase in inflammatory cytokines associated with inflammaging. Additionally, hMMP1 transgenic mice showed a significantly higher susceptibility to developing skin papillomas, underscoring the pivotal role of MMP1 expression in fibroblasts in dermal aging and the creation of a dermal environment conducive to keratinocyte tumor development, a phenomenon often observed in the elderly population [67].

In essence, the collagen-rich dermal microenvironment plays a critical role in shaping the phenotype and behavior of fibroblasts. The age-related decline in fibroblast function can largely be attributed to their adaptation to the degeneration of the collagen-rich ECM microenvironment. These findings introduce a novel concept, suggesting that age-related fibroblast dysfunction primarily reflects adaptive responses to the deteriorating dermal ECM microenvironment, thus establishing an “outside-in adaptation” perspective on the mechanism of dermal aging. This perspective is rooted in the fundamental principle of cell biology, emphasizing the intrinsic connection between form and function. The form and function of dermal fibroblasts are fundamentally dictated by the structure, composition, organization, and mechanical properties of the dermal ECM they inhabit. Therefore, the collagen-rich ECM microenvironment serves not only as a physical scaffold, but also as a critical regulator of fibroblast behavior.

## 3. The Molecular Foundation Underlying the Anti-Aging Properties of Retinol

Numerous approaches have been proposed to hinder the aging process and revitalize aged skin, yet presently, there are no reliable and secure treatments available to counteract skin atrophy in older individuals. Retinol, a derivative of vitamin A, has demonstrated its effectiveness in promoting various anti-aging benefits for the skin and is predominantly favored within the category of retinoids [68,69]. It can stimulate collagen synthesis, inhibit MMP activity, reduce oxidative stress, and modulate gene expression [68,70,71]. Retinol has exhibited efficacy in ameliorating the visual manifestations of both intrinsic and extrinsic aging, such as wrinkles, fine lines, and irregular pigmentation [68,72,73,74]. The mechanisms of retinol action may involve the activation of retinoic acid receptors (RARs) and retinoid X receptors (RXRs), which regulate gene transcription and cell differentiation. Retinol may also modulate the activity of growth factors and cytokines involved in ECM turnover and inflammation. Retinoids, which refer to a group of vitamin A derivatives, are among the most-extensively studied ingredients in skincare for combatting aging and enhancing the appearance of mature skin. Retinoic acid (RA) is the active form of vitamin A and is also called retinol (ROL). ROL serves as a precursor to retinoic acid and can be converted into its active metabolite within human skin. When retinol is applied topically to human skin, it can penetrate the skin and undergo sequential conversion to retinaldehyde and then to retinoic acid [75]. ROL undergoes a two-step conversion process to become retinoic acid. In the first step, the enzyme alcohol dehydrogenase (ADH) catalyzes the conversion of retinol to retinaldehyde, while in the second step, retinaldehyde dehydrogenase (RALDH) converts retinaldehyde to retinoic acid. Studies have demonstrated that vitamin A and its metabolites can enhance the condition of skin that has aged due to both chronological factors and sun exposure by stimulating the formation of new collagen and preventing its breakdown [68,71]. Topical ROL has shown remarkable anti-skin aging effects, suggesting that ROL is a promising and safe anti-aging natural product.

The precise cellular and molecular mechanisms underlying the anti-aging benefits of retinoic acid are still not entirely understood. Additionally, when compared to retinoic acid, there is a scarcity of research investigating the molecular foundation of the anti-aging effects of topical retinol in human skin in live subjects. Understanding the molecular mechanisms through which retinoids enhance aging in human skin has been difficult because of the lack of appropriate in vitro models. The binding of retinoic acid to the retinoid receptors exerts its biological actions. Cells within both the epidermal and dermal layers possess the complete complement of proteins and receptors essential for the physiological actions of vitamin A metabolites in the skin. Nevertheless, when primary keratinocytes or dermal fibroblasts are cultivated as monolayers, their reactivity to retinoic acid treatment is modest. This restricted responsiveness can be attributed, in part, to the diminished presence of nuclear retinoid receptors, which are instrumental in modulating the expression of genes under the control of retinoids. Skin-equivalent cultures present a promising avenue for investigating the regulatory role of retinoids in collagen homeostasis. These cultures feature stratified and differentiated keratinocytes, representing the epidermal layer, layered atop a collagen lattice primarily comprising Type I collagen. Dermal fibroblasts are embedded within this lattice to mimic the dermal layer. When subjected to retinoic acid treatment, these skin-equivalent cultures exhibit a substantial increase in the number of keratinocyte layers and elicit a dermal response akin to the effects observed when retinoic acid is topically applied to human skin in vivo [76]. Consequently, skin-equivalent cultures hold significant potential as a valuable model for delving into the mechanisms by which retinoids enhance the appearance of aging skin in humans.

### 3.1. Increasing the Thickness of the Epidermis and the Vascularity of the Dermis in Aged Human Skin In Vivo Using Topical ROL: Stimulating the Growth of Epidermal Keratinocytes and Dermal Endothelial Cells

The application of topical ROL to aged human skin in a live setting has been found to significantly enhance the thickness of the epidermis by stimulating the proliferation of epidermal keratinocytes [71,77]. Moreover, in addition to improving epidermal thickness, topical ROL has shown a notable increase in the proliferation of endothelial cells and blood vessels in the papillary dermis [71,78,79]. These findings suggest that the topical application of ROL results in the thickening of the epidermal layer and the development of fresh blood vessels within the dermis. This occurs through the stimulation of the growth of epidermal keratinocytes and dermal endothelial cells in the context of aging human skin in vivo. The AP-1 transcription factor is of paramount importance in fostering the proliferation of keratinocytes in response to growth factors, cytokines, and various stimuli [1]. Since the AP-1 complex consists of c-Jun and c-Fos, it has been observed that topical ROL significantly increases the expression of the epidermal-specific c-Jun protein, leading to a substantial increase in epidermal thickness. In contrast, there has been no observed change in the expression of c-Fos protein with topical ROL treatment. These findings suggest that topical ROL enhances the activity of the epidermal-specific c-Jun transcription factor, thereby stimulating the proliferation of epidermal keratinocytes in aged human skin in vivo.

### 3.2. Topical ROL Improves the Dermal ECM Microenvironment by Promoting the Production of Collagenous ECM in Aged Human Skin In Vivo

Topical ROL treatment increases Type I collagen expression, the major structural protein in the skin [71]. Besides the increase in Type I collagen, the application of topical ROL significantly enhances the expression of fibronectin and tropoelastin. In aged human skin in vivo, topical ROL effectively activates dermal fibroblasts, leading to the substantial production of collagenous ECM through the activation of the TGF-β/Smad pathway, which is the principal regulator of ECM production. Topical ROL administration causes a significant increase in TGF-β1 mRNA expression and a decrease in inhibitory Smad7, while other components of the TGF-β pathway remain unaffected. Additionally, topical ROL leads to an increase in the expression of connective tissue growth factor (CTGF/CCN2), which is substantially reduced in the dermis of aged individuals and contributes to the decline in collagen production associated with aging. These findings indicate that topical ROL stimulates the production of ECM by dermal fibroblasts through the upregulation of the TGF-β/CTGF pathway in aged human skin.

In addition to the upregulation of TGF-β/CTGF pathway, retinoic acid significantly reduces CCN1 gene expression in both naturally aged and photoaged human skin in vivo [76]. CCN1 is a negative regulator of collagen homeostasis by inhibiting the TGF-β/CTGF pathway and stimulating MMPs’ induction [22,32]. These data suggest that the mechanism by which topical ROL improves aged skin, through increased collagen production and inhibition of MMPs, may involve the downregulation of CCN1.

In aging skin, decreased vascularity and thinning of the epidermis are substantial factors contributing to skin fragility and hindered wound healing. Therefore, topical ROL not only enhances ECM production, but also improves the dermal microenvironment by promoting the expansion of vasculature through endothelial cell proliferation in aged human skin. An age-related reduction in cutaneous vasculature has been reported [80,81]. The increased vascularity of the dermis induced by topical ROL can improve skin blood flow and create a more-favorable microenvironment for the homeostasis of the epidermis and dermis. Furthermore, the promotion of epidermal keratinocyte proliferation and the restoration of ECM production by topical ROL could create a supportive environment for the growth of endothelial cells and the development of dermal blood vessels. Epidermal keratinocytes are a significant source of vascular endothelial growth factor (VEGF), a powerful factor in promoting angiogenesis [81]. Furthermore, increased production of dermal ECM has been demonstrated to stimulate the proliferation of endothelial cells. As a result, the augmented dermal vascularity facilitated by ROL may have a significant impact on the homeostasis of both the epidermis and dermis.

### 3.3. Topical ROL Improves Hyperpigmentation

Hyperpigmentation is a common dermatological concern affecting individuals of all skin types and backgrounds. Hyperpigmentation, characterized by the darkening of certain areas of the skin due to excess melanin production, is a common cosmetic concern that affects a wide range of individuals [82]. This condition is often associated with sun damage, post-inflammatory hyperpigmentation, melasma, and age-related changes.

While there are numerous treatment options available, retinoic acid has emerged as a potent and versatile agent in the fight against this aesthetic concern [83,84,85]. Retinoic acid functions primarily through its influence on gene expression and cellular differentiation, impacting melanogenesis and melanin distribution [86,87,88]. The mechanisms by which retinoic acid improves hyperpigmentation include the following: the regulation of melanin production: retinoic acid downregulates the activity of tyrosinase, a key enzyme in melanin synthesis, reducing the production of melanin and its transfer to keratinocytes; the acceleration of cellular turnover: by promoting skin cell turnover, retinoic acid helps to exfoliate pigmented cells, revealing fresh, less-pigmented skin beneath; the inhibition of inflammation: retinoic acid has anti-inflammatory properties, making it effective in addressing post-inflammatory hyperpigmentation; collagen stimulation: retinoic acid can improve skin texture, helping to mitigate the appearance of pigmented lesions and fostering an overall youthful complexion. As such, retinoic acid has emerged as a powerful ally in the battle against hyperpigmentation.

### 3.4. Side Effects and Precautions

The retinoid family consists of both naturally occurring and synthetic analogues of retinol [89]. While it can be highly effective, it is essential to be aware of potential side effects and to take necessary precautions when using retinoic acid. Topical ROL offers a significant advantage over retinoic acid in terms of a reduced occurrence of side effects, including erythema, scaling, dryness, and itching [90,91]. While a significantly higher concentration of ROL (0.4%) is required to attain similar outcomes as observed with topical retinoic acid, ROL triggers identical histological alterations (epidermal thickening and dermal ECM production) as retinoic acid, all without causing retinoid irritation, which is the most-prevalent side effect associated with retinoids. On the other hand, inappropriate or excessive use of topical ROL may also result in potential side effects. These commonly include skin dryness, redness, and peeling, which can cause discomfort. However, these side effects typically diminish over time as the skin adjusts to the product. Contrasted with topical ROL treatment, the consumption of excessive amounts of oral retinol can lead to detrimental health consequences, including symptoms like nausea, vomiting, headaches, and dizziness (LD_50_ 1.5 g/kg in mouse) [92]. There are several common retinol derivatives used in skin topical treatments, including retinyl palmitate, retinol, retinaldehyde, adapalene, tretinoin, and tazarotene [89]. Retinyl palmitate and retinol are considered milder forms of retinoids, while retinaldehyde, adapalene, tretinoin, and tazarotene are stronger and more effective in treating skin concerns such as acne, deep wrinkles, and hyperpigmentation. It is important to note that retinol and tretinoin are among the most-well-known topical applications with strong clinical evidence of their anti-aging properties [89].

### 3.5. Summary

Figure 3 presents the fundamental molecular mechanisms responsible for the anti-aging effects of retinol in aging human skin. The topical application of ROL to aged human skin leads to remarkable transformations in both the epidermis and dermis, predominantly influencing three key categories of skin cells: epidermal keratinocytes, dermal fibroblasts, and endothelial cells. Topical ROL exerts a substantial influence on the thickness of the epidermis by activating IFE stem cells and fostering the proliferation of epidermal keratinocytes. This process involves the activation of the c-Jun transcription factor, which plays a crucial role in driving keratinocyte proliferation. In addition to the effects on the epidermis, topical ROL significantly improves the microenvironment of the dermal ECM by activating fibroblasts and stimulating the formation of dermal blood vessels through endothelial cell proliferation. From a mechanistic perspective, the application of topical ROL triggers the activation of the TGF-β/CTGF pathway, a pivotal regulator of ECM equilibrium. This activation results in an increased deposition of mature collagen within aging human skin. Moreover, the augmented dermal vascularization induced by topical ROL fosters improved circulation in the skin, thereby establishing a more-conducive microenvironment for both the proliferation of epidermal keratinocytes and the activation of dermal fibroblasts. Furthermore, the proliferation of epidermal keratinocytes may also contribute to the development of dermal blood vessels by promoting the expression of VEGF. We propose that the interconnected relationships among keratinocyte proliferation, endothelial cell growth, and dermal fibroblast activation create a mutually reinforcing environment, which may account for the remarkable anti-aging effects of retinol in aging human skin.

## 4. Future Perspective

Understanding these mechanisms is important for developing interventions to slow down the aging process and improve skin health. Numerous strategies are available to mitigate the progression of skin aging, encompassing the application of sunscreen for UV radiation protection, adherence to a nutritious diet, and the utilization of topical products infused with antioxidants, retinoids, and natural ingredients to enhance skin health and enhance its appearance. Several possible avenues for future research in the realm of developing novel interventions for skin aging exist: (1) Natural compounds like polyphenols, flavonoids, and carotenoids have been proposed as anti-aging agents for the skin. Further research could scrutinize these compounds to identify their mechanisms of action, safety profiles, and potential effectiveness as anti-aging agents. (2) Using natural compounds in conjunction with other interventions like sunscreens or retinoids could amplify their effects. Future research should examine the synergies of natural compounds with other interventions and identify optimal combinations for enhancing skin health. Exploring the use of natural compounds in combination with other interventions: Natural compounds may be used in combination with other interventions, such as sunscreens or retinoids, to enhance their effects. Future studies could investigate the synergistic effects of natural compounds in combination with other interventions and determine the optimal combinations for improving skin health, (3) Research indicates that natural compounds could influence the skin microbiome, which is critical to skin health. Further studies could explore the impact of natural compounds on the microbiome and assess the potential of targeting it with natural compounds to enhance skin health. (4) Developing innovative formulations and delivery systems for natural compounds is crucial to improve their efficacy, as poor absorption or stability can limit their effectiveness. Future research should explore novel approaches to enhance the stability, absorption, and efficacy of natural compounds through new formulations and delivery systems. Overall, these are a few potential directions for future research on natural compounds and their effects on the skin. By continuing to investigate the potential of natural compounds for improving skin health, we may be able to develop new and effective interventions for skin aging, photoprotection, and other skin-related concerns.

## Figures and Tables

**Figure 1 biomolecules-13-01614-f001:**
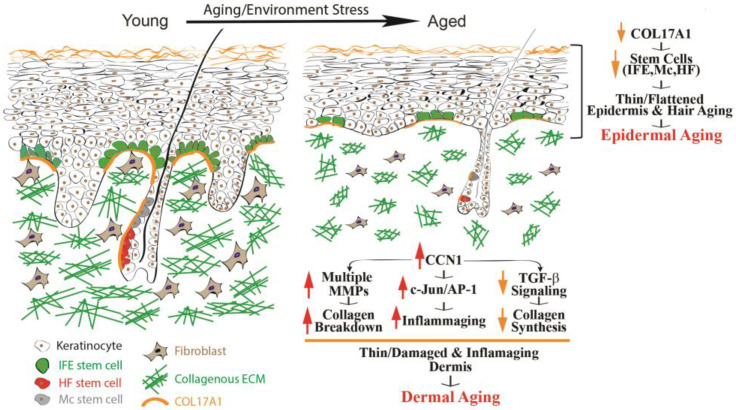
Human skin epidermal and dermal aging. Aging impacts human skin profoundly, with significant changes in its structure. Notable aspects of skin aging encompass the thinning of both the epidermis and dermis, as well as the graying of hair and reduced hair density. Epidermal aging is characterized by diminished expression of COL17A1 in the skin stem cell niche, affecting various components such as the interfollicular epidermis (IFE), melanocyte stem cells (McSCs), and hair follicle stem cells (HFSCs). This reduction weakens the attachment of stem cells to the basement membrane, ultimately contributing to epidermal aging. Meanwhile, dermal aging stems from dysfunction in dermal fibroblasts, including upregulation of CCN1, which drives elevated expression of MMPs, leading to collagen degradation, an increased presence of proinflammatory cytokines that foster an inflammatory microenvironment (referred to as “inflammaging”), and a reduction in collagen production by impairing TGF-β signaling.

**Figure 2 biomolecules-13-01614-f002:**
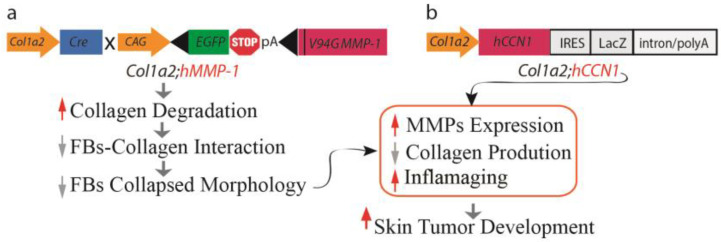
Mouse models of dermal aging. Illustration outlining the establishment of mouse models simulating dermal aging through fibroblast-specific expression of (**a**) catalytically active human MMP1 (*Col1a2;hMMP1*) and (**b**) human CCN1 (h *Col1a2;hCCN1*) under the regulation of the fibroblast-specific *Col-1a2* promoter and enhancer. The fibroblast-specific expression of hMMP1 or hCCN1 leads to the degradation of collagen fibrils, disrupting interactions between fibroblasts and collagen. This disruption results in diminished cell spreading and subsequent adaptive functional changes, contributing to the perpetuation of dermal collagen degradation through the induction of multiple MMPs. Additionally, it inhibits collagen synthesis by interfering with TGF-β signaling and promotes inflammaging by triggering the production of various age-related proinflammatory cytokines. Consequently, dermal aging creates a microenvironment conducive to the development of skin tumors.

**Figure 3 biomolecules-13-01614-f003:**
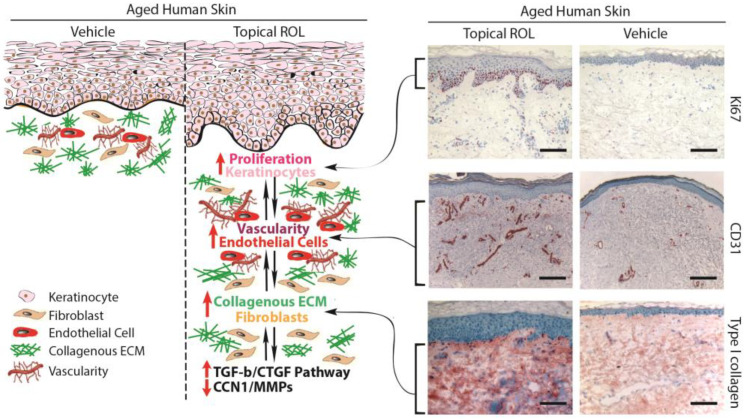
Topical ROL exerts anti-aging effects in aged human skin. Topical ROL triggers changes in the epidermis and dermis, impacting key skin cell types: epidermal keratinocytes, dermal fibroblasts, and endothelial cells. It boosts epidermal thickness by activating IFE stem cells and promoting keratinocyte growth. ROL also enhances the dermal microenvironment by activating fibroblasts, fostering dermal blood vessel formation through endothelial cell proliferation. This occurs via the TGF-β/CTGF pathway, increasing mature collagen in aged skin. The improved vasculature from ROL enhances blood flow, benefiting keratinocyte growth and fibroblast activation. Keratinocyte proliferation might further stimulate VEGF, possibly aiding vessel growth. This interplay between keratinocytes, endothelial cells, and fibroblasts creates a reinforcing environment, potentially explaining ROL’s impactful anti-aging effects in aged skin. Skin sections embedded in OCT (7 μm thickness) were acquired from healthy buttock skin of aged individuals (with an average age of 76 ± 6 years), after a seven-day topical treatment with both a vehicle and 0.4% retinol. Skin sections were immunostained with Ki67 (upper, proliferation marker), CD31 (middle, endothelial cell marker), and Type I procollagen (lower, major collagen in the skin). Representative images of six individuals. Bar = 100 µm.

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
