# Peer review of "Human Skin Aging and the Anti-Aging Properties of Retinol"

_biomolecules, 2023, doi:10.3390/biom13111614_

Round 1

Reviewer 1 Report

Comments and Suggestions for Authors

Human Skin Aging and the Anti-Aging Properties of Retinol by Taihao Quan

In this paper, the author presents main mechanisms of skin aging in humans and some anti-aging properties of Retinol. It is my impression that the paper is written in (too) simplistic way, good for beginners in the field, but may not be suitable for the journal of Biomolecules rank. The main concern regarding this work is that it does not point out any significant new development or novelty in the field of skin aging research, and all mentioned metabolic pathways are presented in very rudimental fashion. However, it might make good introductory reading for a beginner in the field and as such suitable for broader audience.

Certainly, “intrinsic aging” (genetically controlled) cannot be simply attributed to the undefined term "passage of time" and more specific molecular mechanisms must be mentioned and briefly described.

Also, the description below Figure 1 is inadequate, it is too long and it’s just a repetition of the description of skin aging given in the main text.

In line 329 ROL is misspelled as RO):

In section 3., two sentences (“Furthermore, the promotion of epidermal keratinocyte proliferation and the restoration of ECM production by topical ROL could create a supportive environment for the growth of endothelial cells and the development of dermal blood vessels. Epidermal keratinocytes are a significant source of vascular endothelial growth factor (VEGF), a powerful factor in promoting angiogenesis [[39]”.) are repeated twice: between lines 375 – 384.

[[39] in line 379 (or 384!) should be corrected to [39].

A more detailed description and illustration of all mentioned metabolic pathways and interactions would be very welcome.

Reviewer 2 Report

Comments and Suggestions for Authors

Thank you for submitting your manuscript to BIOMOLECULES. I interestingly read your manuscript and have some comments, as follows:

1. In Fig. 2, how can you figure out the difference between dermal and epidermal aging? Are there any specific molecular biology and cell biology-based (biomarker) events in the skin?

2. In the future perspective section, please put your idea on autophagy in the skin.

Comments on the Quality of English Language

Thank you for submitting your manuscript to BIOMOLECULES. I interestingly read your manuscript and have some comments, as follows:

1. In Fig. 2, how can you figure out the difference between dermal and epidermal aging? Are there any specific molecular biology and cell biology-based (biomarker) events in the skin?

2. In the future perspective section, please put your idea on autophagy in the skin.

Round 2

Reviewer 1 Report

Comments and Suggestions for Authors

In section 3.1. title "ROL):" should be corrected to "ROL". Also, there are few "ref." that need to be removed and/or reference introduced.

Author Response

I would like to extend my gratitude to the reviewers for their time.   

The manuscript has been corrected in accordance with the Reviewer’s suggestions.

Changes made to the manuscript are marked in the text.

Point-by-point responses to reviewer’s comments are below.

Reviewer 1

In section 3.1. title "ROL):" should be corrected to "ROL". Also, there are few "ref." that need to be removed and/or reference introduced.

Response: The typographical error has been rectified, and the references have been updated accordingly.

Reviewer 2 Report

Comments and Suggestions for Authors

The revision has been corrected satisfactorily. Thanks. 

Comments on the Quality of English Language

The revision has been corrected satisfactorily. Thanks. 

Author Response

I would like to extend my gratitude to the reviewer's time and effort.